# Detection of Debonding Defects in Concrete-Filled Steel Tubes Using Fluctuation Analysis Method

**DOI:** 10.3390/s24248222

**Published:** 2024-12-23

**Authors:** Chenfei Wang, Yixin Yang, Guangming Fan, Junyin Lian, Fangjian Chen

**Affiliations:** 1College of Civil Engineering, Inner Mongolia University of Science and Technology, Baotou 014010, China; cfwang@xmut.edu.cn; 2College of Civil Engineering and Architecture, Xiamen University of Technology, Xiamen 361024, China; 19065495506@163.com (G.F.); 18960027152@163.com (J.L.); fj_chen@xmut.edu.cn (F.C.); 3Engineering Research Center of Structure Crack Control for Major Project, Fujian Province University, Xiamen 361024, China; 4Inner Mongolia Autonomous Region Building Structure Disaster Prevention and Mitigation Engineering Technology Research Center, Baotou 014010, China

**Keywords:** concrete-filled steel tube, debonding defects, piezoelectric sensors, fluctuation analysis, wavelet packet analysis

## Abstract

This study presents a comprehensive method for detecting debonding defects in concrete-filled steel tube (CFST) structures using wave propagation analysis with externally attached piezoelectric ceramic sensors. Experimental tests and numerical simulations were conducted to evaluate the sensitivity and accuracy of two measurement techniques—the flat and oblique measurement methods—in detecting debonding defects of varying lengths and heights. The results demonstrate that the flat measurement method excels in detecting debonding height, while the oblique method is more effective for detecting debonding length. A normalized judgment index (DI) was introduced to quantify the correlation between debonding characteristics and the detected signal amplitude, revealing the significant influence of sensor spacing on detection accuracy. Furthermore, a mathematical model based on wavelet packet energy analysis was developed to establish a linear relationship between wavelet packet energy and debonding size. This model offers a scientific foundation for the quantitative detection of debonding defects and provides a new approach to the health monitoring of CFST structures. The integrated use of both measurement techniques enhances detection precision, enabling both qualitative and quantitative defect analysis, which can significantly guide the maintenance and repair of CFST structures.

## 1. Introduction

Concrete-filled steel tube (CFST) composite structures are a foundational choice in modern civil engineering due to their robust structural design [1], where concrete is confined within a steel tube [2]. This configuration imposes a triaxial stress state on the concrete, significantly enhancing its load-bearing capacity through the “hoop effect” [3]. As a result, CFST structures exhibit not only superior load-bearing performance but also high seismic resilience and construction efficiency, making them ideal for high-rise buildings, bridges, tunnels, and marine applications. However, debonding defects at the steel–concrete interface can compromise these advantages by reducing the effective contact area, thus lowering the overall load-bearing capacity of the structure. Such defects also lead to local stiffness reductions, resulting in increased deformation and potential displacement under load. Moreover, stress concentrations in areas adjacent to the debonding defects can accelerate material degradation, increasing the likelihood of localized damage and crack propagation under service conditions. Given these potential risks, effective methods for detecting debonding defects in CFST structures are essential to ensure their long-term performance and structural safety [4].

Despite their advantages, CFST structures are vulnerable to interfacial debonding between the steel tube and concrete. This issue can arise due to construction quality inconsistencies, material degradation, or external influences like natural disasters [5]. If left undetected, these defects may compromise the load-bearing integrity and stability of the structure, potentially leading to catastrophic failure with severe consequences for safety and property [6,7]. Consequently, the rapid and precise detection of debonding defects is essential for maintaining the long-term reliability of CFST structures [8,9].

Existing standards, such as the “Technical Standard for Inspection of Building Structures” (GB/T 50344-2019) [10], include various debonding detection methods for CFST structures: the tapping method, core drilling verification, and ultrasonic testing. However, each of these methods has notable limitations. The tapping method heavily depends on the inspector’s subjective experience, reducing its reliability. The core drilling verification method, though accurate, is intrusive and risks damaging the structure. While ultrasonic testing is non-destructive, it is often slow and technically complex, limiting its practical use for large-scale applications. These constraints highlight a critical gap in CFST testing methodologies: the need for a novel, efficient, and non-destructive approach to quickly and accurately detect interfacial debonding across extensive areas. Addressing this gap is not only crucial for advancing CFST structural integrity monitoring but also represents a significant opportunity for innovation in non-destructive testing (NDT) technologies.

The fluctuation analysis method, an active health monitoring technique that utilizes piezoelectric sensors, has emerged as a promising approach for structural defect detection [11]. This dynamic technique leverages fluctuation analysis to offer several significant advantages: it is straightforward to implement, cost-effective, and efficient, while also requiring minimal surface preparation of the target structure [12]. For CFST structures, debonding defects notably influence the propagation characteristics of stress waves, altering the wave patterns that can be detected by sensor systems [13,14]. In recent years, non-destructive testing (NDT) techniques based on the capture and analysis of stress waves using externally mounted piezoelectric ceramic sensors have received increasing research attention [15]. These approaches are gaining traction because they facilitate real-time, in situ monitoring without compromising structural integrity. Moreover, they offer potential advancements in the detection sensitivity and diagnostic capability for interfacial defects, providing a new avenue for rapid and precise assessment of CFST structural health.

Kang et al. [16] analyzed the signal curves of specimens with defects by comparing them with those of healthy specimens, measuring the signal amplitude under different anchorage lengths. Xu et al. [17] conducted finite element analysis to establish a coupled model of piezoelectric materials and CFST structures, revealing the mechanism of wavelet packet analysis based on the response of piezoelectric materials in detecting interfacial debonding in CFST structures. Ramalho [18] proposed a damage detection method based on artificial neural networks using Lamb wave data, which can independently determine the degree of damage in single-lap joints, regardless of the damage location. Reza [19] analyzed the effects of damage size and orientation on Lamb wave scattering behavior through experiments and numerical simulations, optimizing the damage detection process using Lamb waves and providing a basis for damage detection in isotropic plates. Ng [20] used the interaction of Rayleigh waves at the debonding interface of CFRP and concrete as one of the indicators for detecting debonding and proposed a damage reconstruction algorithm for detecting and locating debonding phenomena in CFRP-reinforced concrete structures. Shao et al. [21] demonstrated that when Rayleigh waves encounter underground voids, diffraction occurs, and the energy of the diffracted waves accumulates at the boundaries of the voids, which can be used to determine the location of the voids.

While these studies present effective non-destructive testing methods, limitations remain in their scope and detection precision. Although techniques like Lamb and Rayleigh waves are useful for identifying interface defects, environmental complexities in real-world applications present challenges that have yet to be fully addressed. These constraints underscore the necessity for further development of highly sensitive and robust non-destructive detection methods, particularly for complex CFST interfaces. This study aims to bridge these gaps by introducing innovative detection techniques and algorithms to improve accuracy and reliability in identifying interfacial debonding in CFST structures.

This study aims to advance detection technology for debonding defects in concrete-filled steel tube (CFST) structures through the application of the fluctuation analysis method, specifically targeting the evaluation of debonding post-concrete casting. By systematically investigating the propagation mechanisms of stress waves within CFST structures, this research provides a detailed analysis of how varying debonding lengths and heights affect stress wave behavior. The study not only optimizes existing testing methodologies but also proposes novel approaches for comprehensive assessment. Through the development of robust feature extraction and defect quantification techniques, this work introduces new indices for evaluating debonding and rigorously validates these methods via laboratory experiments. Additionally, numerical simulations shed light on underlying mechanisms, while wavelet packet energy analysis quantifies the impact of debonding dimensions on energy distribution. These findings contribute valuable scientific insights and experimental data to inform the health monitoring and maintenance strategies of CFST structures, supporting enhanced structural integrity and safety.

## 2. Detection Principle of Debonding Defects in Concrete-Filled Steel Tubes Based on the Fluctuation Analysis Method

### 2.1. The Piezoelectric Effect

The piezoelectric effect refers to the phenomenon in which certain materials generate electric charge when subjected to mechanical stress or deformation or undergo mechanical deformation when subjected to an external electric field. This effect includes both the direct piezoelectric effect and the inverse piezoelectric effect. The direct piezoelectric effect involves the generation of electric charge in response to mechanical stress or deformation, while the inverse piezoelectric effect involves mechanical deformation in response to an applied electric field [22]. In practical applications, both the direct and inverse piezoelectric effects are often considered simultaneously, as Figure 1. The coupling equations are given by:(1)$$D_{i}=d_{ijk}T_{jk}+\in_{ij}E_{j}$$
(2)$$S_{jk}=s_{jklm}T_{jk}+d_{ijk}E_{i}$$
where *D_i_* represents the electric displacement component (i.e., charge density), *d_ijk_* is the piezoelectric strain constant, *T_jk_* represents the stress tensor component, *S_jk_* represents the strain tensor component, *E_i_* is the electric field intensity component, $\in_{ij}$ is the dielectric constant, and $s_{jklm}$ is the elastic constant.

In this study, the fundamental principles of the piezoelectric effect are employed to detect debonding defects within concrete-filled steel tubes using piezoelectric ceramic sensors. This approach effectively captures internal structural information, providing data support for subsequent qualitative and quantitative analysis.

### 2.2. Mechanism of Stress Wave Propagation in Concrete-Filled Steel Tubes

When detecting debonding defects in concrete-filled steel tubes, selecting the appropriate type of wave based on the fluctuation analysis method is crucial for accurate defect detection and assessment. This study primarily utilizes surface waves, specifically analyzing the propagation characteristics of Lamb waves and Rayleigh waves and their application in both flat and oblique measurement methods.

#### 2.2.1. Selection of Surface Waves

Lamb waves are elastic waves that propagate in solid thin plates [23], with their wave characteristics influenced by both the plate thickness and the frequency. Lamb waves are known for their sensitivity to structural defects, energy concentration, and long propagation distances, making them widely used in non-destructive testing. Rayleigh waves are elastic waves that propagate along the surface of a solid [24], with their energy concentrated near the surface and rapidly attenuating with depth. Rayleigh waves are particularly sensitive to surface defects, making them suitable for detecting surface and near-surface defects. The propagation equations for Lamb waves and Rayleigh waves are as follows:(3)$${(k}_{T}^{2}-k_{L}^{2})\;\sin(k_{T}d)\;\cos(k_{L}d)+4k_{T}k_{L}\sin(k_{T}d)\;\cos(k_{L}d)=0$$
(4)$${\left(\frac{V_{R}}{V_{S}}\right)}^{2}=\frac{0.87+1.12\nu }{1+\nu }$$
where $k_{L}$ and $k_{T}$ represent the wave numbers for longitudinal and transverse waves, respectively, and d is the thickness of the steel tube wall. $V_{R}$ denotes the velocity of Rayleigh waves, $V_{S}\;$is the shear wave velocity, and ν is the Poisson’s ratio.

#### 2.2.2. Propagation and Interaction of Waves in Flat and Oblique Measurement Methods

In the flat measurement method, piezoelectric ceramic sensors are placed parallel to the surface of the concrete-filled steel tube components, primarily utilizing Lamb waves for detection. As stress waves propagate along the surface, they encounter debonding defects, which cause scattering, reflection, and mode conversion, resulting in noticeable changes in the received waveform [25]. The flat measurement method provides a relatively intuitive reflection of surface and shallow-layer defect information. During the propagation of stress waves, interactions with debonding defects are primarily manifested as reflection, scattering, and mode conversion. Some stress waves are reflected back to the sensor upon encountering a defect, leading to changes in the amplitude of the received signal. Simultaneously, multiple-direction scattering waves are generated at the defect site, resulting in the received signal containing additional frequency components and noise. Moreover, different modes of Lamb waves may convert at the defect site, affecting the spectral characteristics of the received signal.

In the oblique measurement method, piezoelectric ceramic sensors are positioned at an angle to excite and receive stress waves through oblique incidence. This method primarily utilizes Rayleigh waves for detection. The interaction of stress waves with debonding defects during propagation is more complex in this case. The oblique-incidence stress waves are refracted at the defect site, altering the wave’s propagation path and velocity. Some stress waves are also reflected by the defect; however, the angle of incidence introduces greater complexity to the direction of the reflected waves. Additionally, in the oblique measurement method, scattering waves include not only surface scattering but also internal scattering, resulting in more diverse received signals. The mode conversion phenomenon is more pronounced due to the oblique incidence angle, providing additional spectral information about the defects.

By analyzing the interactions between stress waves and defects in both the flat and oblique measurement methods, a more comprehensive assessment of the health of concrete-filled steel tube structures can be achieved, providing accurate results for defect detection and evaluation, as Figure 2.

### 2.3. Qualitative Analysis of Debonding Defects Based on Normalized Judgment Index

In the detection of debonding defects in concrete-filled steel tube (CFST) structures, the normalized judgment index (DI) is widely used to assess the health of the structure. The flat measurement method and oblique measurement method detect debonding defects through different stress wave propagation characteristics. By integrating these two methods, a more comprehensive qualitative analysis of debonding defects can be achieved. The flat measurement method relies on the propagation characteristics of Lamb waves, which are particularly sensitive to debonding height due to their propagation speed and attenuation characteristics in CFST structures. When using the flat measurement method, the amplitude and phase information of the signal are significantly affected by the debonding height, leading to notable variations in the calculated DI value as the debonding height increases. Therefore, the flat measurement method is particularly suitable for identifying and evaluating the height of debonding defects. The normalized DI value can quickly determine whether there are debonding defects in the structure and provide an estimate of their extent. On the other hand, the oblique measurement method relies on Rayleigh waves. Rayleigh waves propagate along the surface and near-surface regions of the CFST structure, exhibiting high sensitivity to surface and shallow-layer defects. Although the fluctuations in DI values during qualitative detection with Rayleigh waves are not as pronounced as with Lamb waves, their sensitivity to surface defects, combined with wavelet packet energy analysis, provides valuable information for further quantitative detection. Analyzing the attenuation and changes in the signal during Rayleigh wave propagation offers more comprehensive information for structural health assessment.
(5)$$DI=|\frac{D_{i}-D_{P}}{D_{P}}|$$
where *Di* represents the amplitude of the sine signal received by the externally attached piezoelectric ceramic sensor, $D_{P}$ denotes the amplitude of the sine signal received by the piezoelectric ceramic sensor when the specimen is intact, and *i* is the test label.

To overcome the limitations of individual detection methods, this study proposes a comprehensive detection method based on the combination of the flat measurement method and the oblique measurement method. By integrating the advantages of both methods and using the normalized judgment index (DI), the combined approach provides a holistic assessment of the impact of debonding defects on structural health. Specifically, the flat measurement method, through the analysis of Lamb waves, yields DI values related to debonding height, while the oblique measurement method, utilizing Rayleigh waves’ sensitivity to surface and near-surface defects, enhances the detection capability for debonding defects. The integration of these two methods allows not only for the identification of the presence of debonding defects but also provides a solid foundation for subsequent quantitative analysis. By combining the flat and oblique measurement methods, a comprehensive inspection of internal defects from different angles is achieved, avoiding the potential blind spots of a single method. Additionally, the use of the normalized judgment index (DI) across different detection methods significantly improves defect detection sensitivity, particularly in accurately reflecting the true health state of the structure when addressing complex debonding defects.

In summary, the comprehensive detection method proposed in this study effectively enables qualitative analysis of debonding defects in concrete-filled steel tube structures and provides a reliable and practical technical means for structural health monitoring.

### 2.4. Quantitative Analysis of Debonding Defects Based on Wavelet Packet Energy

In the quantitative analysis of debonding defects in concrete-filled steel tubes, wavelet packet energy analysis offers an effective tool for exploring the propagation characteristics of stress waves within the structure and their interaction with defects. To analyze detailed information in the experimental signals, this study employs a wavelet packet decomposition method with three levels. This approach performs multi-scale analysis on the signals obtained during testing, enabling the extraction of wavelet packet energy across various sub-frequency bands. This multi-scale analysis method helps capture the energy distribution characteristics of different frequency components, allowing for more precise identification and quantification of the size of debonding defects.

#### 2.4.1. Wavelet Packet Analysis of Different Waveforms

Lamb waves, commonly used for detecting thin-walled structures, exhibit significant variations in propagation characteristics across different frequencies. In this study, the energy distribution of Lamb waves in wavelet packet analysis reveals a combined effect of high-frequency and low-frequency components. However, due to the presence of debonding defects, the energy in the high-frequency range is significantly enhanced. Debonding defects cause local changes in structural stiffness, leading to more pronounced scattering and reflection of high-frequency components as Lamb waves traverse the defect area. Rayleigh waves primarily propagate along the surface and near-surface regions of the structure. Their propagation characteristics make them more sensitive to surface and shallow defects. In wavelet packet energy analysis, Rayleigh waves exhibit notable changes in low-frequency and mid-frequency components. Particularly in the presence of debonding defects, the energy in these frequency bands is significantly increased. This enhancement is attributed to local scattering and attenuation of the waveform caused by the defects, which results in increased energy in the low-frequency and mid-frequency ranges of Rayleigh waves. The wavelet packet energy can be computed from the experimental signals using the following equation:(6)$$E_{j,k}=\sum_{n}|w_{j,k}(n){|}^{2}$$
where *E_j,k_* denotes the wavelet packet energy value at the *k*-th node of the *j*-th layer, *w_j,k_*(*n*) represents the coefficient at the *k*-th node of the *j*-th layer at time n, and n denotes the time index sample point.

#### 2.4.2. Establishment of Fitting Formulas and Quantitative Analysis

Based on the wavelet packet energy obtained from the analysis, fitting formulas related to debonding length and height are further derived. By analyzing the variations in wavelet packet energy under different debonding lengths and heights, a quantitative relationship model can be established. Initially, wavelet packet energy is extracted from numerical simulation data corresponding to various debonding lengths and heights. Multivariate regression analysis is then employed to model the relationship between wavelet packet energy and debonding length and height. Using the fitted formula, quantitative analysis of debonding defects in steel tube concrete can be conducted. Indoor experimental signal data is analyzed using wavelet packet analysis to compute the wavelet packet energy in each sub-band. The calculated wavelet packet energy is then substituted into the fitted formula to estimate the corresponding debonding length and height. The accuracy and effectiveness of the fitting formula are validated by comparing the actual defect sizes with the results obtained from the fitting. This wavelet packet energy-based quantitative analysis method enables precise evaluation of debonding defects by extracting frequency domain features from the signals, providing an effective means for health monitoring of steel tube concrete structures.

## 3. Experimental Study on Steel Tube Concrete with Debonding Defects

### 3.1. Design of Steel Tube Concrete Specimens

In this experiment, seven Q235 square steel tube concrete specimens were designed and cast, with various sizes of artificially simulated defects adhered to the inner wall. Each specimen featured a cross-sectional dimension of 400 × 400 mm^2^ and a height of 50 mm, with a steel tube wall thickness of 10 mm, and was filled with C50 self-compacting concrete. While this height is less than the cross-sectional dimension and smaller than typical member heights in real-world structures, this design was chosen for experimental feasibility and to ensure consistent and accurate measurements. Specifically, the reduced height allowed for precise control over the placement and size of the debonding defects while minimizing material use and experimental complexity. Despite this limitation, the study maintains structural relevance by focusing on the local behavior of stress waves and their interaction with debonding defects, which is independent of specimen height in this context.

Some researchers [26] have used foam boards to simulate debonding defects; however, due to the tendency of foam boards to deform during concrete casting, this approach was not considered in this study. Given that acrylic plates possess a low elastic modulus and high strength, they effectively simulate the debonding phenomenon between the steel tube and concrete caused by delamination, as Figure 3. In this experiment, acrylic plates were used to simulate debonding defects and were symmetrically attached to the center of the inner wall of the steel tube, as shown in Figure 4. After attachment, the positions of the pre-set defects were marked on the outer surface of the steel tube using a marker pen.

In this study, a total of 12 acrylic plates of varying sizes were sequentially attached to simulate discontinuous debonding defects with debonding areas of 50 × 25 mm^2^, 100 × 25 mm^2^, and 150 × 25 mm^2^, and debonding heights of 5 mm, 10 mm, 15 mm, and 20 mm. These acrylic plates were applied to replicate different debonding lengths and heights. The specimens were tested in groups using both flat measurement and oblique measurement methods and the specimens after casting are shown in Figure 5. The specific specimen labels are detailed in Table 1 below:

### 3.2. Experimental Setup and Testing Methods

#### 3.2.1. Testing Equipment

The experiment utilized a RIGOL DG1022 arbitrary function signal generator (Beijing Puyuan Jingdian Technology Co., Ltd., Beijing, China), a YOKOGAWA DLM2054 oscilloscope (Yokogawa Electric (China) Co., Ltd., Shanghai, China), and a single-transducer piezoelectric ceramic sensor (PXR03RMH) produced by Changsha Pengxiang Company, Changsha, China. The piezoelectric ceramic sensor was equipped with a magnetic ring on its surface to ensure better adhesion to the steel tube wall. The experimental setup involved generating a continuous sine wave signal with a frequency of 20 kHz and an amplitude of 10 V using the signal generator. The piezoelectric used in the experiment is shown in Figure 6. Prior to testing, a coupling agent was evenly applied to both the piezoelectric ceramics and the designated positions on the steel tube wall [27], and the test site is shown in Figure 7.

#### 3.2.2. Testing Methods

##### Flat Measurement Method

In the flat measurement method, piezoelectric sensors were placed on the external surface of the concrete-filled steel tube structure, with their polarization axis aligned parallel to the structure’s surface to ensure effective propagation of stress waves along the surface. To investigate the impact of different sensor spacings on the detection results, two sensor arrangement configurations were designed for the experiment: one with a spacing equal to half of the side length of the steel tube (20 cm) and the other with a spacing equal to three-quarters of the side length (30 cm). The specific testing method is shown in Figure 8.

##### Oblique Measurement Method

In the design of the oblique measurement method, the piezoelectric sensors were installed at the midpoint of the opposite surface of the concrete-filled steel tube structure, with their polarization axis oriented at a certain angle relative to the structure’s surface. This arrangement enhances the interaction between the stress waves and structural defects, thereby improving the accuracy of detecting debonding defects. The specific testing method is shown in Figure 9.

### 3.3. Analysis of Experimental Results

This study thoroughly investigates the different amplitude variation trends exhibited by the planar and oblique measurement methods in detecting debonding defects in concrete-filled steel tube structures. By extracting time-domain signals from the oscilloscope during laboratory tests, the time-domain waveform for the PC30-3 group is illustrated in Figure 10.

Table 2 reveals that the planar and oblique measurement methods exhibit different trends in signal amplitude variation. Specifically, the planar method shows a significant positive correlation between signal amplitude and debonding height. As debonding height increases, the signal amplitude progressively rises, indicating that the planar method is highly sensitive to waveform changes caused by variations in debonding height. In contrast, the oblique measurement method demonstrates a negative correlation; as debonding height increases, the signal amplitude gradually decreases. This observation is further analyzed using the DI values obtained from different methods, as calculated by Equation (5).

#### 3.3.1. Impact of Debonding Height

Through the analysis of different experimental groups, the maximum DI value for the PC20 group reached 1, while the maximum DI value for the PC30 group reached 1.17, as shown in Figure 11. The DI values for the PC30 group were consistently higher than those for the PC20 group, indicating that increasing the sensor spacing can improve detection accuracy. However, in the case of the oblique measurement method, the highest DI value for debonding height was only 0.79, which is significantly lower compared to the planar measurement method, as illustrated in Figure 12. The DI values for the oblique measurement method also showed less variability. Therefore, the oblique measurement method is less effective than the planar measurement method for detecting debonding height. Based on these results, the planar measurement method demonstrates greater sensitivity and accuracy in detecting debonding height and is more suitable for qualitative analysis of debonding defects.

#### 3.3.2. Impact of Debonding Length

When detecting debonding length, the planar measurement method and the oblique measurement method exhibit different characteristics. Although the DI values from the planar measurement method are generally higher than those from the oblique method, the rate of increase in DI values slows as the debonding length increases. This suggests that while the planar method is effective at detecting the presence of debonding defects within the structure, it is less sensitive to changes in debonding length. In practical detection scenarios, the relatively minor changes in DI values with increasing debonding length indicate that the planar method is better suited for detecting features like debonding height, which are closely related to changes in local structural stiffness. In contrast, the oblique measurement method, despite having lower DI values overall, shows a more pronounced response to changes in debonding length. As the debonding length increases from 50 mm to 150 mm, the DI values for the oblique method increase rapidly, with a maximum growth rate of 371%. In comparison, the maximum growth rates for PC20 and PC30 are only 35.4% and 36.3%, respectively. The relationship between DI values and debonding length under different conditions is shown in Figure 13. This phenomenon indicates that the oblique method has greater sensitivity to debonding length, making it more effective in identifying the extent of defects within the structure.

The oblique method leverages the propagation characteristics of Rayleigh waves, which provides a significant advantage in detecting defects that propagate along the structure’s surface. This advantage allows the oblique method to better capture the energy changes caused by defects during wave propagation. Consequently, the rapid increase in DI values can effectively reflect the debonding length, making the oblique method more suitable for qualitative detection of debonding length.

In summary, the oblique measurement method demonstrates certain advantages in detecting debonding length, especially when identifying longer debonding defects. The significant changes in the DI values observed with this method provide a reliable basis for defect analysis. By enhancing sensitivity to variations in debonding length, this detection method offers a more detailed assessment tool for monitoring the health of concrete-filled steel tube structures, complementing the limitations of the planar measurement method in debonding length detection. This approach achieves a more comprehensive defect detection and evaluation. To further elucidate the phenomena observed in the laboratory tests, a stress wave simulation study of CFST was conducted.

## 4. Numerical Simulation

### 4.1. Properties and Mesh

In the established finite element model, the density of C50 concrete was set to 2.4 × 10⁻⁹ t/mm^3^, with an elastic modulus of 3.5 × 10⁴ MPa and a Poisson’s ratio of 0.2. For Q235 steel, the density was set to 7.8 × 10⁻⁹ t/mm^3^, with an elastic modulus of 2.06 × 10⁵ MPa and a Poisson’s ratio of 0.3. Both the steel and concrete were assigned a mass damping coefficient of 0.1 and a stiffness damping coefficient of 0.001. The density of the piezoelectric ceramics was set to 7.75 × 10⁻⁹ t/mm^3^, with elastic properties defined using engineering constants. Additionally, orthotropic dielectric constants and strain-type piezoelectric material properties were specified, with a stiffness damping coefficient of 0.03, and the material direction was assigned as 3. To accurately simulate and capture the propagation of stress waves, the mesh size was set to less than one-tenth of the wavelength, as calculated using Equation (3). The mesh size for the concrete was set to a maximum of 19 mm, and for the steel, it was set to a maximum of 25.5 mm. To ensure precise simulation of stress wave transmission within the concrete-filled steel tube structure, the concrete mesh was refined to 10 mm, and the mesh size in the debonding defect area was further refined to 5 mm, with section types assigned as C3D10. The steel mesh was also set to 10 mm, with section types assigned as S4R, while the piezoelectric ceramic sections were assigned as C3D8E. The specific mesh layout is illustrated in Figure 14. After completing the mesh setup, the specimens were assembled. To enhance simulation efficiency, a configuration of one transmitter and two receivers of piezoelectric ceramics was arranged on the steel tube wall surface, allowing for both flat and oblique measurement methods to be employed in the piezoelectric testing [28,29].
(7)$$\lambda =\sqrt{\frac{E}{\rho }}\times \frac{1}{f}$$
where λ represents the wavelength of the stress wave, E is the elastic modulus, ρ denotes the material density, and f is the signal frequency.

### 4.2. Steps and Interactions

In the analysis steps, Implicit Dynamics was chosen, with the total time set to 1 × 10⁻^3^ s. Geometric nonlinearity was disabled, the maximum number of increments was set to 10,000, and the increment size was fixed at 1 × 10⁻⁶. Additionally, the EPOT (electric potential) field output was included, with the frequency adjusted to 1. The interfaces between the steel and concrete were set as tie constraints, as well as between the piezoelectric ceramic and the steel. Given that the elastic modulus of the steel is significantly higher than that of both the concrete and the piezoelectric ceramic, the steel was designated as the master surface, while the contact surfaces of the concrete and piezoelectric ceramic were set as the slave surfaces. To more accurately capture the electric potential received by the piezoelectric ceramics, tie constraints were also applied to the piezoelectric sensors on both the flat and oblique measurement sides.

### 4.3. Load

A continuous sine wave signal with a frequency of 20 kHz and an amplitude of 10 V was applied to the top surface of the piezoelectric ceramic actuator, with the bottom surface set to 0 V. For the piezoelectric ceramic sensors, the bottom surface potential was also set to 0 V, and nodes were placed on the top surface to receive the electric potential. To accurately simulate the propagation of stress waves within the concrete-filled steel tube, complete constraints were applied to the bottom of the cross-sectional model of the steel tube and concrete specimen. For the flat measurement method, constraints were applied to restrict movement in the *y*-axis direction on the surface of the piezoelectric ceramics. For the oblique measurement method, constraints were applied in the *x*-axis direction on the surface of the piezoelectric ceramics.

### 4.4. Visualization

The stress distribution diagrams illustrate the propagation of stress waves within the concrete-filled steel tube, as shown in Figure 15. The completed model was used to output field variables, plotting the electric potential on the top surfaces of the piezoelectric ceramics for both parallel and oblique arrangements. This allowed for the assessment of electric potential energy using both the flat and oblique measurement methods on the steel tube and concrete cross-sectional components. For example, Figure 16 shows the time-domain signal simulated for the PC30-3 group. The maximum values of the obtained electric potentials were used to compute the normalized judgment index (DI) using Formula (5). The maximum difference in DI values for the flat measurement method was 1.08, while for the oblique measurement method, the maximum DI difference was 0.83. Furthermore, numerical simulations were performed on concrete-filled steel tube specimens with various debonding defects. The time-domain plots derived from the electric potential values in the field variable outputs were used to calculate the normalized judgment index (DI), as illustrated in Figure 17. By comparing the DI values obtained from numerical simulations and indoor experiments for both the flat and oblique measurement methods—where the numerical simulation DI values are marked with (M)—Figure 18 reveals that the trends in DI values from the numerical simulations are consistent with those from the indoor experiments.

The normalized judgment index (DI) obtained through numerical simulations aligns closely with the results of the laboratory experiments. To further investigate the relationship between debonding height, debonding length, and the corresponding test signals, a wavelet packet energy analysis was conducted on the time-domain signals derived from both numerical simulations and laboratory tests. This analysis provides deeper insights into how variations in defect characteristics influence the energy distribution within the signals, thereby offering a more comprehensive understanding of the defect detection process in concrete-filled steel tube structures.

## 5. Analysis of Debonding Evaluation Indicators

Wavelet packet analysis not only combines the advantages of Fourier transform and wavelet analysis but also excels in decomposing both low-frequency and high-frequency components of signals [30,31]. In this study, wavelet packet decomposition and reconstruction were performed on the measured signals of steel tube concrete specimens with varying degrees of debonding [32]. MATLABR2020a software was used to calculate the wavelet packet energy of the received signals. By comparing the wavelet packet energy from specimens with different debonding defect volumes to that of intact specimens, the relationship between debonding defects and wavelet packet energy measured through wave propagation was established. Using debonding length and height as variables, linear fitting of the wavelet packet energy obtained from numerical simulations was conducted to derive fitting formulas. These formulas were then applied to wavelet packet energy values obtained from three testing methods in the indoor experiments, yielding R^2^ values that indicate the goodness of fit of the regression models. This quantitative relationship between debonding length, debonding height, and wavelet packet energy provides a means for the quantitative detection of debonding length and height in concrete-filled steel tubes. The calculated wavelet packet energy results for the signals from the PC20, PC30, and XC groups are shown in Figure 19.

The correlation coefficient R^2^ of the fitting curve for the PC20 group is 0.9304, which is lower than the R^2^ value of 0.9432 for the PC30 group. This suggests that in the flat measurement method, increasing the distance between the piezoelectric ceramic sensor driver and the sensor improves the detection of debonding defects. Additionally, the XC group, with a correlation coefficient R^2^ of 0.9775, shows even better fitting results when using the oblique measurement method for the quantitative identification of debonding defects in concrete-filled steel tubes. Consequently, the relationship between the wavelet packet energy of the signals measured by sensors using different testing methods and the debonding length and height in concrete-filled steel tubes can be expressed as follows:y_pc20_ = 0.1175 − 0.0003x_l_ + 0.0062x_h_ − 0.0002x_h_^2^(8)
y_pc30_ = 0.1052 − 0.0006x_l_ + 0.0079x_h_ − 0.0003x_h_^2^(9)
y_xc_ = 0.1026 − 0.0004x_l_ + 0.0012x_h_ − 0.0001x_h_^2^(10)
where x_l_ represents the debonding length; x_h_ represents the debonding height; y_pc20_ denotes the wavelet packet energy value obtained using the flat measurement method with a 20 cm distance between the transmitter and receiver; y_pc30_ represents the wavelet packet energy value obtained using the flat measurement method with a 30 cm distance between the transmitter and receiver; and y_xc_ denotes the wavelet packet energy value obtained using the oblique measurement method.

## 6. Conclusions

This study proposes a comprehensive testing method using wave propagation analysis with externally attached piezoelectric ceramic sensors to detect debonding defects in concrete-filled steel tube (CFST) structures. Experimental tests and numerical simulations validated the effectiveness of this method in identifying defects of various sizes. The following conclusions were drawn:Stress waves in CFST structures primarily propagate along the surface of the steel tube when debonding occurs. The flat measurement method exhibits a positive correlation between debonding and output voltage amplitude, while the oblique method shows a negative correlation. The normalized judgment index (DI) quantifies these relationships, with increasing sensor distance enhancing detection accuracy in the flat measurement method.The flat measurement method is more sensitive to debonding height, while the oblique measurement method is better suited to detecting debonding length, offering advantages in quantitative analysis.A linear relationship between wavelet packet energy and debonding length and height was established, providing a basis for the quantitative detection of defects. This model enhances detection accuracy and quantifies the extent of damage.

In summary, the combined use of flat and oblique measurement techniques offers a more comprehensive and accurate method for detecting and quantifying debonding defects in CFST structures. This approach improves detection precision and provides a robust framework for the health monitoring and maintenance of CFST structures.

## Figures and Tables

**Figure 1 sensors-24-08222-f001:**
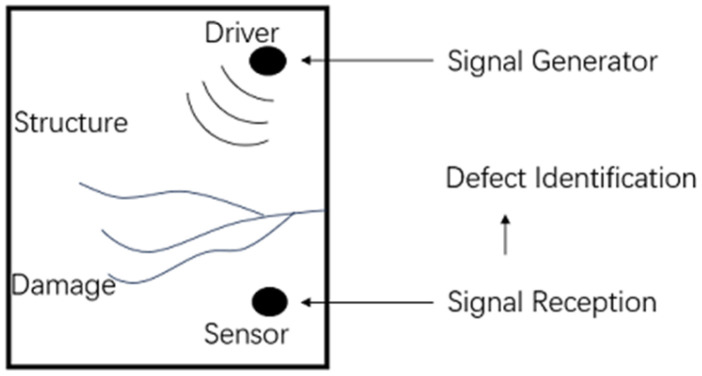
Piezoelectric effect-based detection principle.

**Figure 2 sensors-24-08222-f002:**
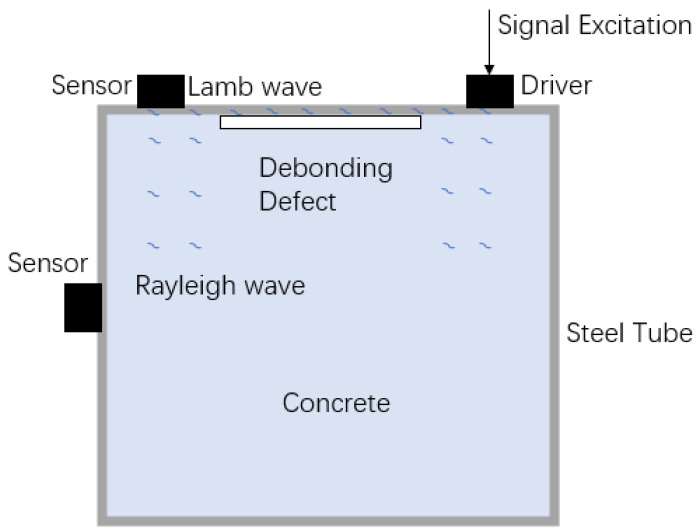
Internal stress wave propagation in concrete-filled steel tubes.

**Figure 3 sensors-24-08222-f003:**
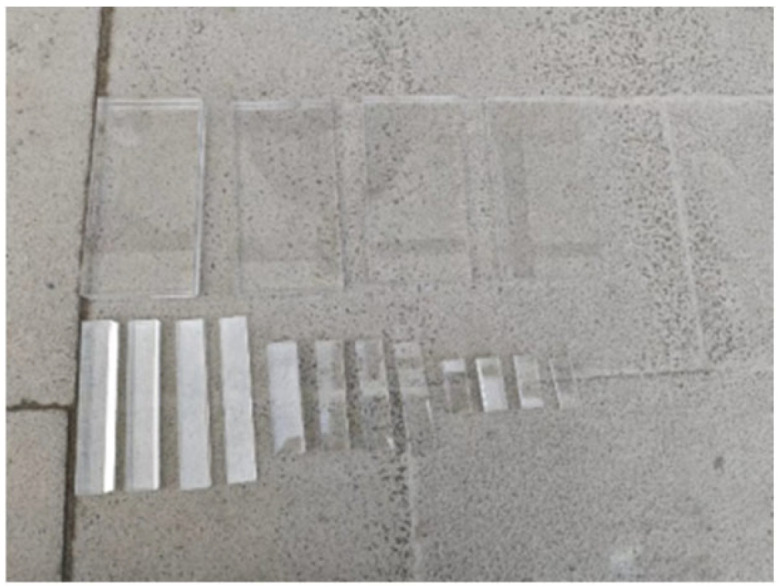
Acrylic plate used in the experiment.

**Figure 4 sensors-24-08222-f004:**
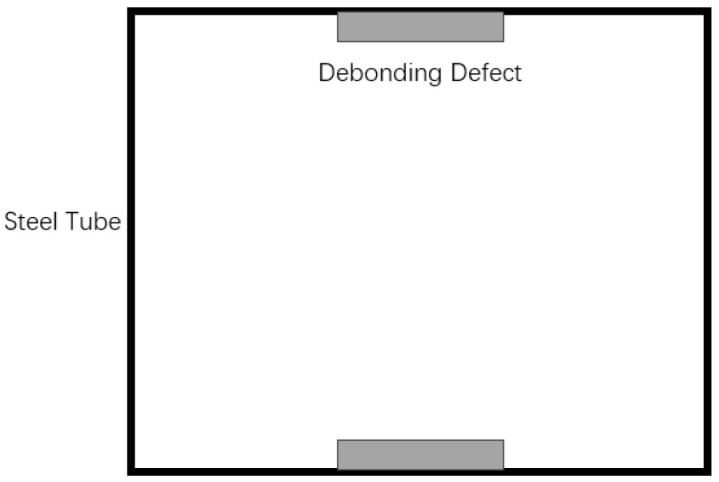
Schematic of defect attachment.

**Figure 5 sensors-24-08222-f005:**
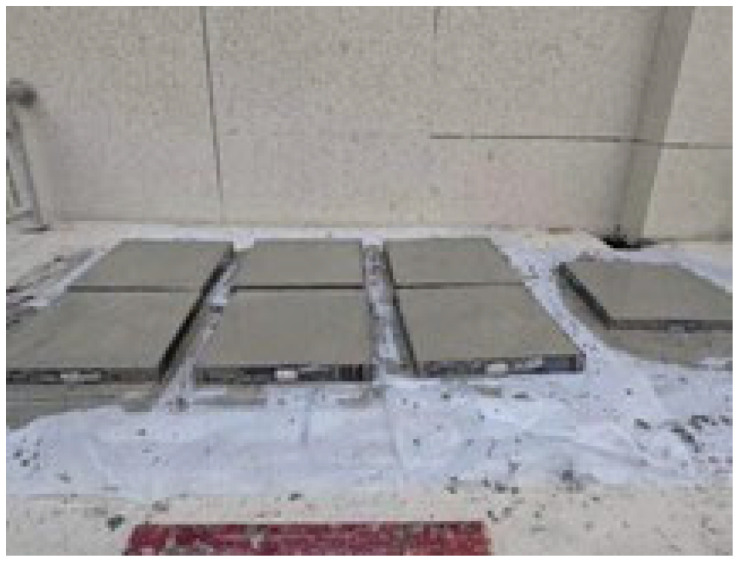
Concrete-filled steel tube specimen after pouring.

**Figure 6 sensors-24-08222-f006:**
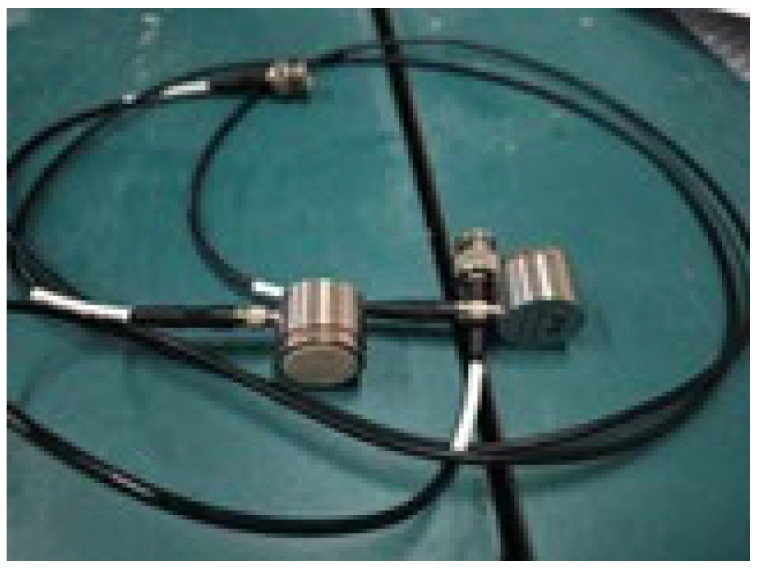
Piezoelectric ceramic sensor with magnetic attachment.

**Figure 7 sensors-24-08222-f007:**
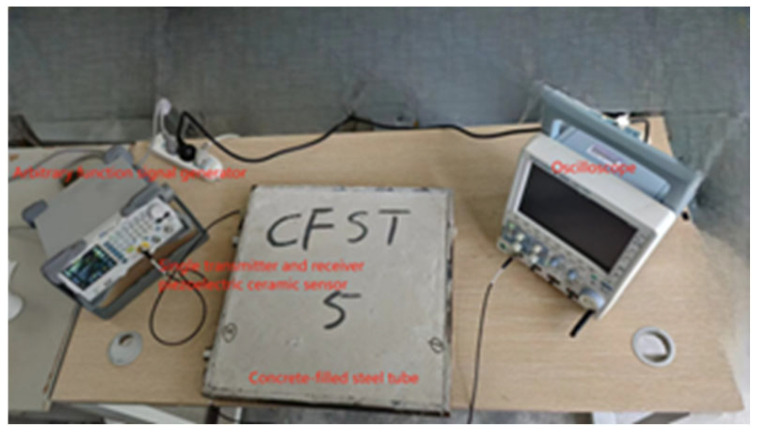
Diagram of external piezoelectric ceramic sensor testing.

**Figure 8 sensors-24-08222-f008:**
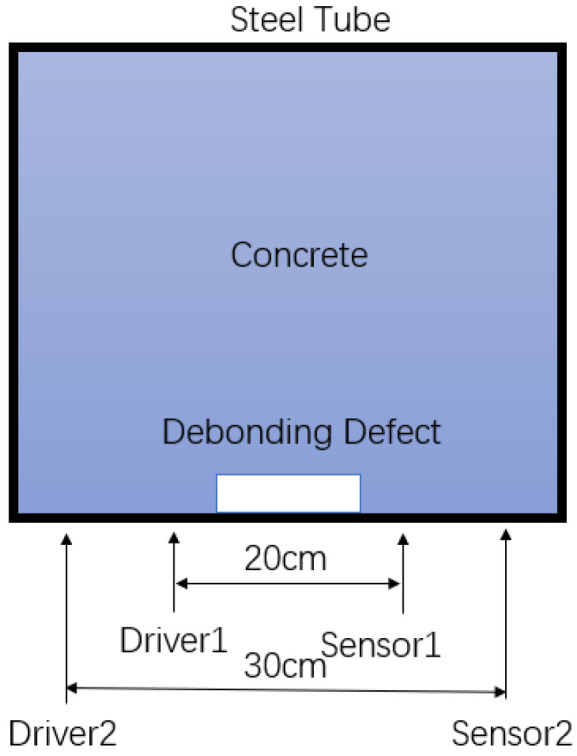
Flat measurement method measurement point diagram.

**Figure 9 sensors-24-08222-f009:**
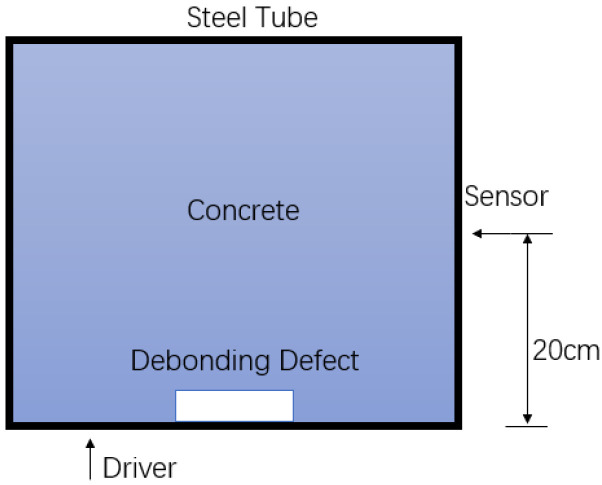
Oblique measurement method measurement point diagram.

**Figure 10 sensors-24-08222-f010:**
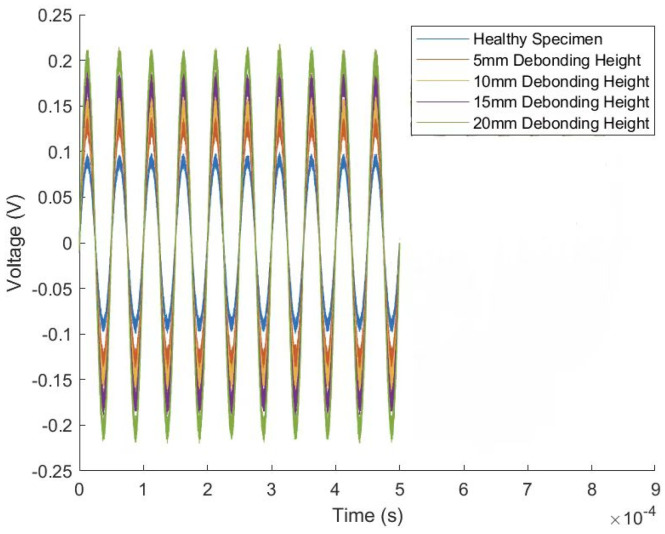
Time-domain plot for PC30-3 group.

**Figure 11 sensors-24-08222-f011:**
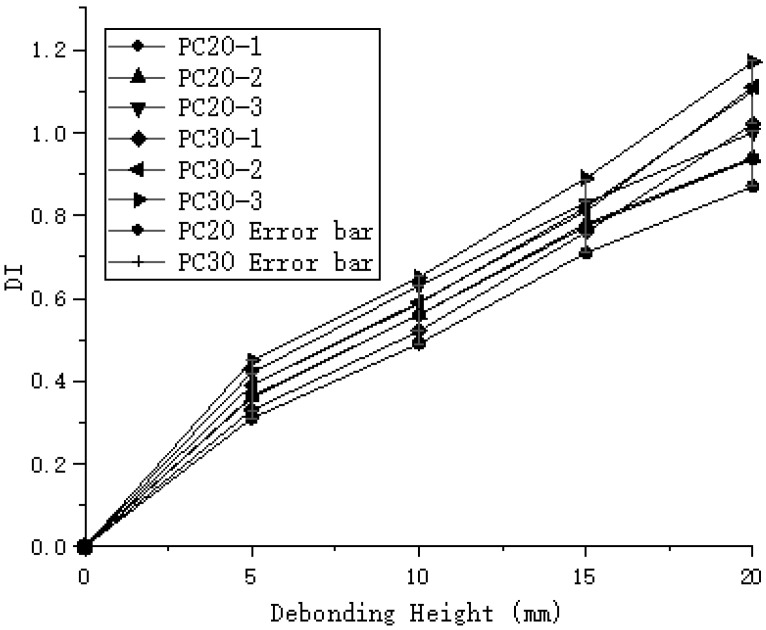
Normalized Defect Index (DI) for planar measurement method.

**Figure 12 sensors-24-08222-f012:**
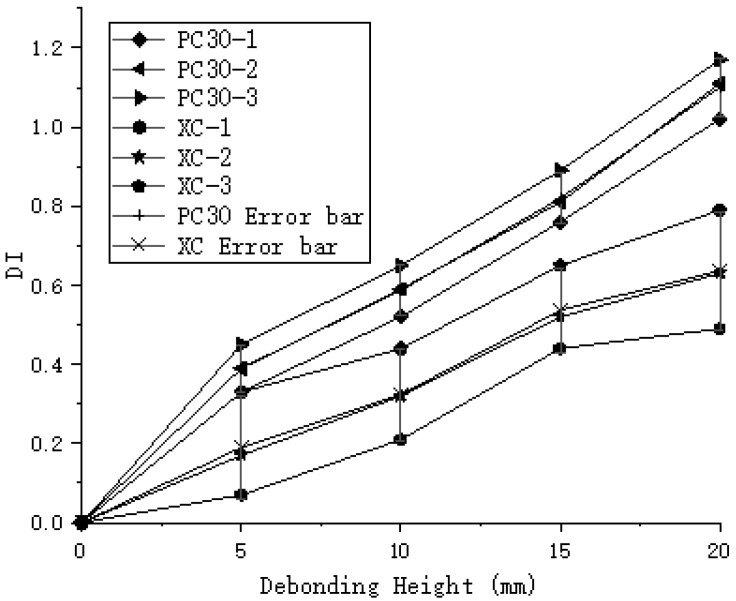
Comparison of DI between oblique and planar measurement methods.

**Figure 13 sensors-24-08222-f013:**
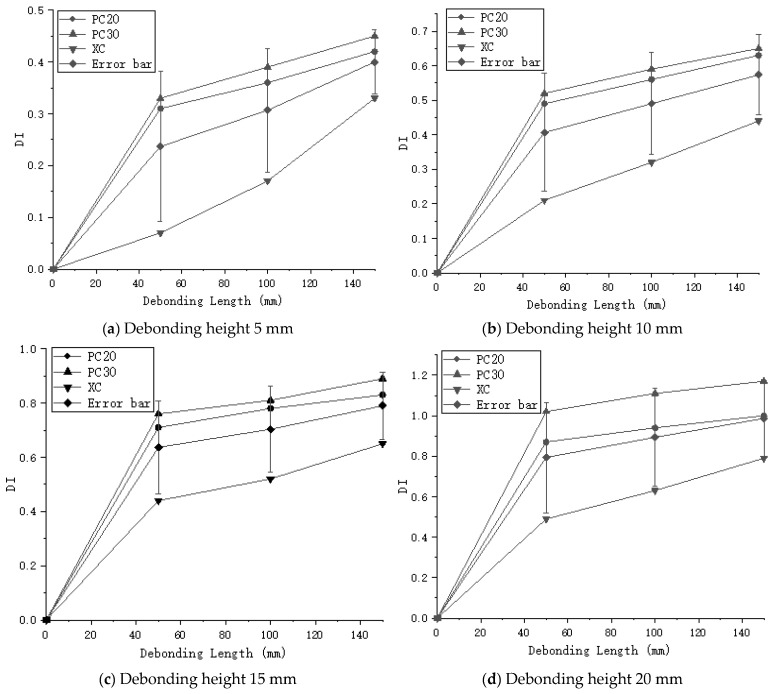
Comparison of DI between oblique and planar measurement methods based on debonding length at different debonding heights.

**Figure 14 sensors-24-08222-f014:**
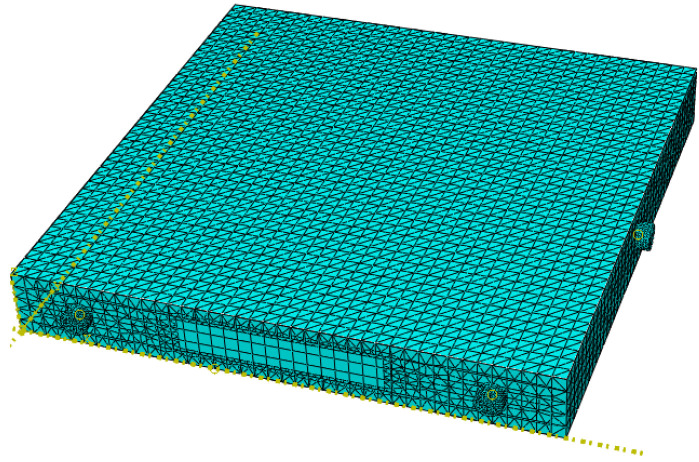
Mesh division of concrete-filled steel tube specimens.

**Figure 15 sensors-24-08222-f015:**
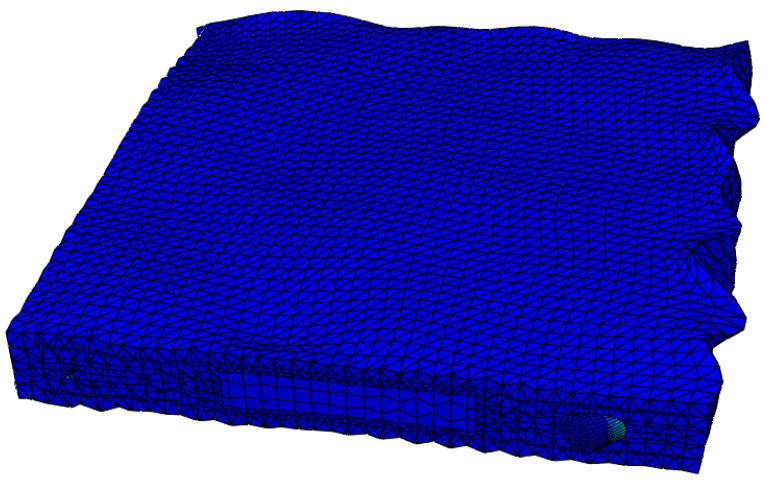
Stress contour map from numerical simulation.

**Figure 16 sensors-24-08222-f016:**
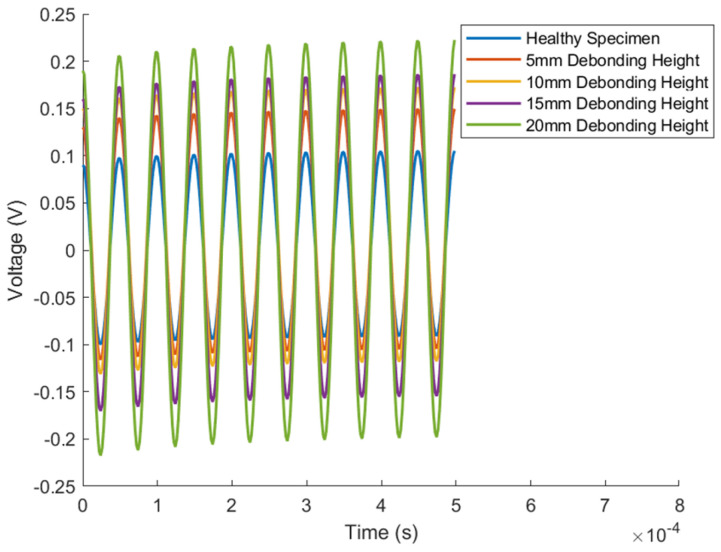
Time-domain signal for PC30-3 from numerical simulation.

**Figure 17 sensors-24-08222-f017:**
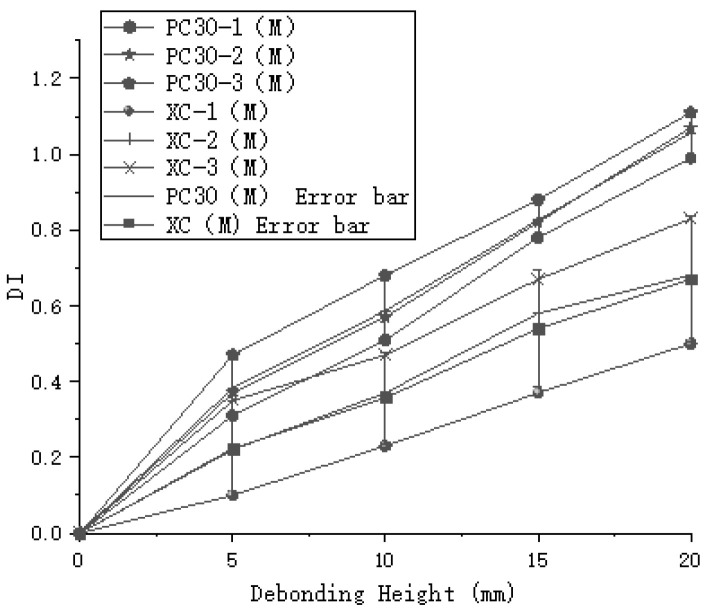
Normalized judgment index (DI) from numerical simulation.

**Figure 18 sensors-24-08222-f018:**
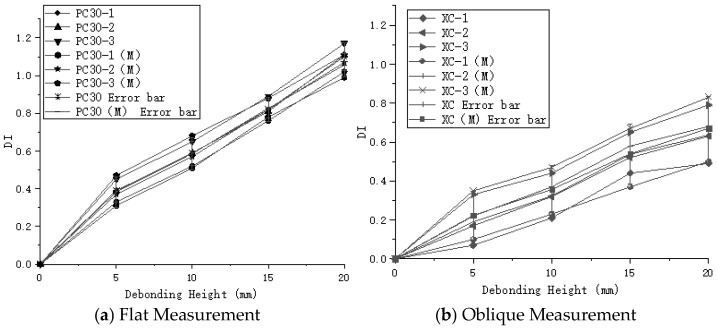
Comparison of normalized judgment index (DI) between indoor experiments and numerical simulation.

**Figure 19 sensors-24-08222-f019:**
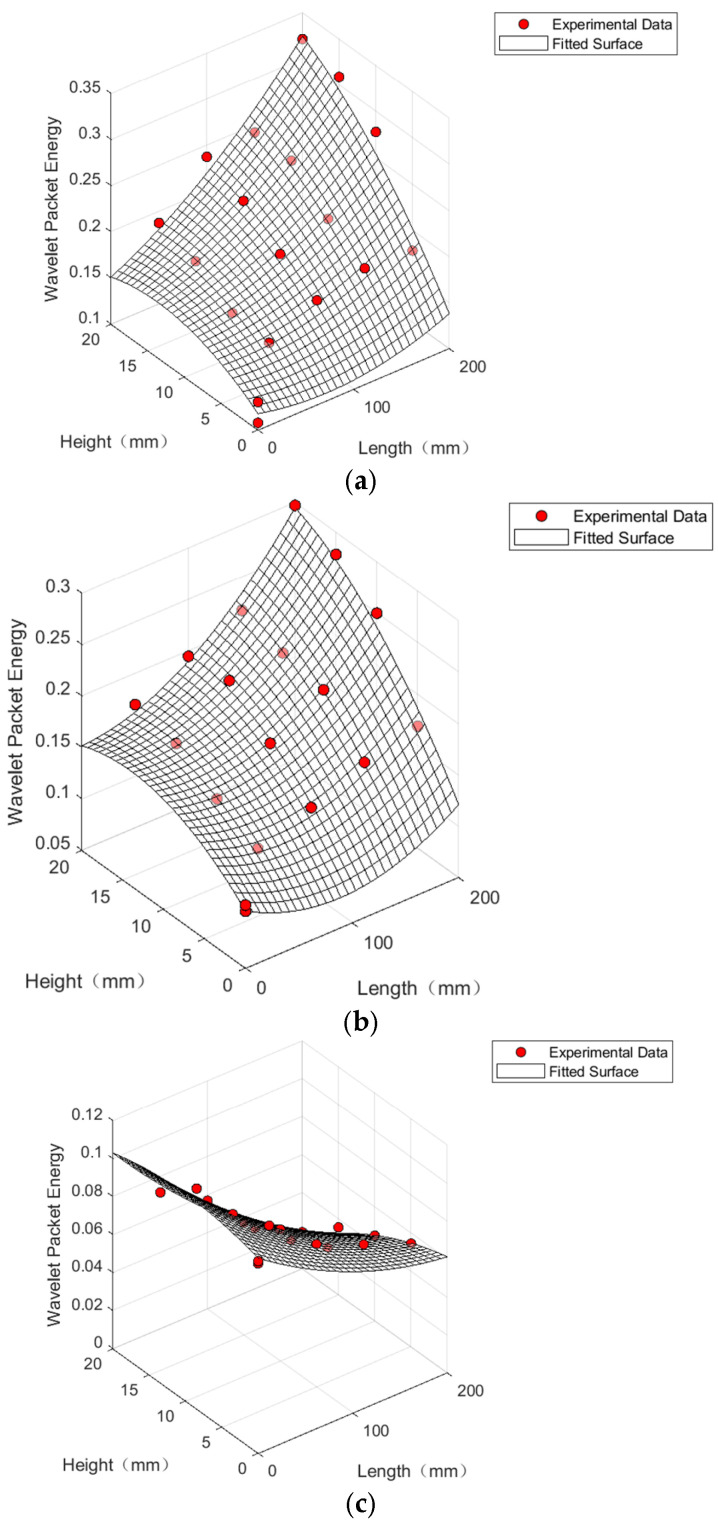
Evaluation indicators for different testing methods under various defect conditions. (**a**) Evaluation indicators for PC20 at different levels of debonding. (**b**) Evaluation indicators for PC20 at different levels of debonding. (**c**) Evaluation indicators for PC20 at different levels of debonding.

**Table 1 sensors-24-08222-t001:** Data from different testing methods.

Serial Number	Debonding Area (a × b) (mm^2^)	Debonding Height (mm)	Serial Number	Debonding Area (a × b) (mm^2^)	Debonding Height (mm)
PC20-1	50 × 25	5, 10, 15, 20	PC30-3	150 × 25	5, 10, 15, 20
PC20-2	100 × 25	5, 10, 15, 20	XC-1	50 × 25	5, 10, 15, 20
PC20-3	150 × 25	5, 10, 15, 20	XC-2	100 × 25	5, 10, 15, 20
PC30-1	50 × 25	5, 10, 15, 20	XC-3	150 × 25	5, 10, 15, 20
PC30-2	100 × 25	5, 10, 15, 20			

“PC” represents the flat measurement method, “XC” represents the oblique measurement method, and “20” and “30” denote the spacing between the transmitting and receiving piezoelectric ceramic sensors.

**Table 2 sensors-24-08222-t002:** Amplitude values measured by different testing methods.

Serial Number	Measured Amplitude (V) at Different Debonding Heights			
	5 mm	10 mm	15 mm	20 mm
PC20-1	0.147	0.167	0.192	0.209
PC20-2	0.152	0.175	0.199	0.217
PC20-3	0.159	0.183	0.205	0.224
PC30-1	0.133	0.152	0.176	0.202
PC30-2	0.139	0.159	0.181	0.211
PC30-3	0.145	0.165	0.189	0.217
XC-1	2.35 × 10^−2^	1.99 × 10^−2^	1.41 × 10^−2^	1.29 × 10^−2^
XC-2	2.09 × 10^−2^	1.71 × 10^−2^	1.21 × 10^−2^	9.32 × 10^−3^
XC-3	1.68 × 10^−2^	1.41 × 10^−2^	8.80 × 10^−3^	5.30 × 10^−3^

## Data Availability

Data will be made available on request.

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
