# Peer review of "Detection of Debonding Defects in Concrete-Filled Steel Tubes Using Fluctuation Analysis Method"

_sensors, 2024, doi:10.3390/s24248222_

Round 1

Reviewer 1 Report

Comments and Suggestions for Authors

The paper presents the fluctuation analysis method for detecting the debonding defects in concrete-filled steel tubes. The paper is interesting, well-written and structured.

1.      Line 275: “These specimens have cross-sectional dimensions of 400×400×50 mm³, …” Is this cross-sectional or dimension of the prisms? Cross-sectional should be square (400x400 mm2). Is 50 mm the height of specimens? Why did the authors design the height of specimens less than the cross-sectional dimension? In real structures, the member height/span is always much larger than the cross-sectional dimension.   

2.      The figures are blurry. Improve the quality of figures.

3.      How the outcome of this study will benefit researchers and end users? This needs to be highlighted.

4.      Suggest to provide the scientific reason for Conclusion 2 in the Discussion Section.

Author Response

We thank the reviewers for their constructive feedback, which has greatly contributed to improving the quality and clarity of our manuscript. Below, we provide detailed responses to the comments:

Comments 1:Line 275: “These specimens have cross-sectional dimensions of 400×400×50 mm³. Is this cross-sectional or dimension of the prisms? Cross-sectional should be square (400x400 mm2). Is 50 mm the height of specimens? Why did the authors design the height of specimens less than the cross-sectional dimension? In real structures, the member height/span is always much larger than the cross-sectional dimension.

Response 1: Clarification of specimen dimensions and height: We have clarified the dimensions of the specimens in line 290, explicitly specifying the cross-sectional size and height of the specimens. Additionally, in line 294, we have addressed the rationale behind the chosen specimen height, providing a justification for its design in the context of the study.

Comments 2:The figures are blurry. Improve the quality of figures.

Response 2: Replacement of unclear figures: We have carefully reviewed and replaced all unclear figures in the manuscript to enhance visual clarity and ensure they effectively support the text.

Comments 3:How the outcome of this study will benefit researchers and end users? This needs to be highlighted.

Response 3: Enhancement of the conclusion section: The conclusion has been revised to emphasize the research outcomes and highlight the novelty and contributions of the study more effectively.

Comments 4:Suggest to provide the scientific reason for Conclusion 2 in the Discussion Section.

Response 4: Refinement of conclusion points: The conclusion section has been further condensed and reorganized into concise points to enhance clarity and readability.

To facilitate the review process, we have highlighted all modifications made in response to your comments in red within the revised manuscript. We believe these revisions address the reviewers' concerns thoroughly and improve the manuscript significantly. Thank you for your valuable suggestions. 

Reviewer 2 Report

Comments and Suggestions for Authors

Paper consider the problem of debonding defects in CFST. Before further consideration, some points should be improved, namely:

1) Please do not use any abbreviation in the Abstract.

2) Introduction. Please give detailed state of art related to the problem of debonding defects in CFST. Please describe the generation, geometry, prefereable direction and propagation mechanism of such defects. 

3) Figs 11-13, 18. Error bars should be added.

Author Response

We sincerely appreciate the reviewer’s thoughtful feedback and suggestions. The following revisions have been made to address the comments provided: 

Comments 1:Please do not use any abbreviation in the Abstract.

Response 1: Abstract revision: The abstract has been completely rewritten to better condense the article's content, while avoiding the use of abbreviations for improved clarity and accessibility.

Comments 2:Introduction. Please give detailed state of art related to the problem of debonding defects in CFST. Please describe the generation, geometry, prefereable direction and propagation mechanism of such defects. 

Response 2: Enhancement of the introduction: Additional details regarding the characteristics and impact of debonding defects have been incorporated into the introduction. This has led to a revision of the content to provide a more comprehensive foundation for the study. 

Comments 3:Figs 11-13, 18. Error bars should be added.

Response 3: Addition of error bars: Error bars have been added to Figures 11–13 and 17–18 to reflect variability and ensure a clearer presentation of the experimental data.

All modifications made in response to the reviewer’s comments have been highlighted in red within the revised manuscript for ease of review.We hope these changes address the reviewer’s concerns and further enhance the quality of the manuscript. Thank you for your valuable input. 

Round 2

Reviewer 2 Report

Comments and Suggestions for Authors

Paper can be accepted now